# A Systematic Review Exploring the Economic Valuation of Accessing and Using Green and Blue Spaces to Improve Public Health

**DOI:** 10.3390/ijerph17114142

**Published:** 2020-06-10

**Authors:** Mary Lynch, Llinos Haf Spencer, Rhiannon Tudor Edwards

**Affiliations:** 1School of Health Sciences, Bangor University, Bangor LL57 2EF, UK; 2Centre for Health Economics and Medicine Evaluation, Bangor University, Bangor LL57 2PZ, UK; l.spencer@bangor.ac.uk (L.H.S.); r.t.edwards@bangor.ac.uk (R.T.E.)

**Keywords:** valuing nature, public health, physical activity, green spaces, blue spaces, economic evaluation

## Abstract

Contact with the natural environment in green and blue spaces can have a valuable influence on population physical and mental health and wellbeing. The aim of this study is to explore the economic evidence associated with the public’s value for accessing, using and improving local environments to undertake recreational activity and consuming the associated health benefits of green and blue spaces. Applying the Preferred Reporting Items for Systemic Reviews and Meta-Analyses (PRISMA) guidelines, a systematic literature search was conducted. Peer-reviewed articles were sought using electronic databases, scrutiny of reference lists, experts and grey literature. All relevant papers meeting the criteria were critically appraised for methodological quality using the Drummond checklist. The review search concluded with 12 papers applying the Grades of Recommendation, Assessment, Development and Evaluation (GRADE) approach to assess the quality with a narrative analysis conducted under the themes. Results suggest the public value access to green and blue spaces to undertake recreational activities and avoid delay or losing the recreational experience and associated health benefits. The public are willing to pay between £5.72 and £15.64 in 2019 value estimates for not postponing or losing an outdoor experience and for walking in local environments under current and improved environmental conditions, respectively. Valuation estimates indicate the public value green and blue spaces and are willing to pay to improve local environments to gain the health benefits of undertaking leisure activities in green and blue spaces.

## 1. Introduction

Engagement with the natural environments and settings with a range of biodiversity can provide considerable health and wellbeing benefits for the general population [1]. Demand and investment in health improvement are multifactorial and complex. Health is both a consumption and investment good and dependent on the amount of resources an individual allocates to the production of health through inputs such as time and health-improving behaviours [2]. Physical activity (PA) is viewed as an essential health-improving input in the health production process, with PA in green and blue spaces (GABS) affording physical and mental health outcomes [3]. See Appendix A for Glossary of terms. The natural environment encompassing GABS can be defined as green spaces that are settings including vegetation in structured and unstructured environments (e.g., parks and gardens) [4] and blue spaces as accessible settings principally consisting of water (e.g., rivers and lakes) [5]. GABS are resources that are used to promote health and wellbeing through the reduction of stress and risk for poor mental health [6,7,8,9], and its relevance in land-use planning has been advocated [10,11]. Evidence also indicates that visits to GABS can reduce the risk of chronic diseases such as cardiovascular diseases [12], obesity [13] cancer [14] and type 2 diabetes [15]. A lifestyle characterised by even a little PA has been shown to be beneficial to health [16,17]. Therefore, interventions that encourage engaging with GABS can have beneficial influences on promoting health and wellbeing [18].

The access and availability of GABS in local residential environments can promote and encourage participation in PA and supporting active living [19,20], social interaction [21], enhance cognitive development and reduce aggressive behaviours [22,23]. Time spent in GABS can improve self-discipline, promote better health in the elderly and delay the impact of dementia [24].

Contact with nature and involvement in nature-based interventions have been shown to facilitate each of the five ways to wellbeing, which include connecting with others, being active, taking notice, keep learning and giving [1,25,26]. The delivery of health-related therapeutic interventions in natural environments such as wilderness therapy [27] is becoming increasingly popular approach in the treatment of mental health conditions. Worldwide, mental health problems are one of the main causes of overall disease burden [28]. Access to green spaces in local environments can have benefits for mental health and wellbeing [29], with exercise outdoors in natural environments associated with higher wellbeing, along with lower levels of stress and anxiety [7]. Investing in new environmental infrastructures could help with health and wellbeing and result in social and economic gains while saving on healthcare resource uses. Evidence suggests that parks and green spaces are estimated to save the NHS around £111 million per year based only on a reduction in the number of visits to the GP and excluding other costs such as prescriptions and referrals. In addition, the public attributes a value for accessing and using parks and greens spaces, with evidence suggesting that frequent users have a higher willingness to pay (WTP), with value estimates ranging from £2.89 to £3.93 per month in the UK [30]. Individual, social and physical environmental factors all have an interrelated role to play in promoting and increasing levels of PA. Research suggests potential health impacts have a probable cost-effectiveness ratio of £4469 per disability adjusted life year (DALY) [31]. Evidence of changes in the quality of neighbourhood characteristics suggest that improvements to local environments can increase walkability and could be a cost-effective way to increase PA levels, with demands for walkable environments estimated at £13.65 per person per week or £710 per person annually to instigate a policy change [32]. The current available evidence that connects nature with health and wellbeing are not fittingly incorporated into policy developments [24]. The purpose of this systematic review is to examine all relevant economic evidence relating to the economic benefits and impacts of GABS on physical and mental health and wellbeing. This systematic review is a new area for health economic research linking economic evaluation methodologies primarily used in environmental economics and modified to incorporate health and wellbeing effects and reflecting the public’s perceived values. A motivation for this review was to examine the evidence to ascertain if methodological approaches can effectively transverse disciplines to take account of multifactorial problems regarding the investment and allocation of resources promoting health-improving behaviours. The findings could provide a more concise framework for methodological guidance on the benefits of green and blue spaces that are not currently defined with much precision.

## 2. Materials and Methods 

The aim of this systematic review was to determine what economic evidence is available while examining the importance and value of GABS. To scrutinise the evidence, two review questions were developed:(1)What is the economic evidence on the health and wellbeing benefits of green or blue spaces?(2)What are the available standard tools for evaluating nature-based health and wellbeing interventions?

These research questions, along with a flow diagram of the systematic review, are outlined in Figure 1. This systematic review shadows the University of York Centre for Research and Dissemination (York CRD) principles for conducting searches and extracting data [33].

To examine the review questions and search strategy, a PICO (Patient/Problem or Population, Intervention, Comparator and Outcome) framework was used which is a mnemonic used in evidence-based practice to frame and answer a health care related question [34]. This framework facilitated the construction of the search process: search terms directly from the review objectives by defining and focusing on the key attributes of the review topic (see Appendix B). Advise was also sought from a systematic review specialist and by looking at the search terms used by other researchers [35,36,37].

Databases used for the evidence search strategy were the Cochrane Collaboration Register and Library, CINAHL, ASSIA, PsycINFO, PubMed incorporating MEDLINE, Web of Science, Database of Abstracts of Reviews of Effects (DARE), NHS Economic Evaluation Database (EED) and HTA. Screening of reference lists and hand-searching were used to supplement and add to the electronic searching. Grey literature was included to limit publication bias using online search engines. Owing to inadequate translation resources, only articles written or translated into English (UK and international) were eligible for inclusion. Search terms and keywords are a mixture of MESH (Medical Subject Heading) and non-MESH terms (see Appendix C). To ensure that the correct articles were identified, search terms were divided into 3 groups: population, intervention and outcomes, and an information scientist was consulted to help shape the search terms and truncate keywords. Search terms were linked with “or” Boolean operators within groups and with “and” Boolean operators between groups. The literature was searched from 1988 to January 2018, as the size of the published scientific literature has expanded exponentially over the last 30 years [38]. This was to ensure that older and newer evidence was captured on valuing nature. 

The inclusion criteria were articles containing components of economic evaluation of contact with nature, connecting with others in nature, being active in nature, taking notice of nature, learning in nature and giving in nature in either GABS [1,25,26]. The exclusion criteria were all health and wellbeing interventions that did not involve economic evaluation nor contact with nature, connecting with others in nature, being active in nature, taking notice of nature, learning in nature and giving in nature in either blue spaces or green spaces [1,28,29].

The initial screening process was conducted by two of the authors (M.L. and L.H.S.). Articles were evaluated for relevance against the eligibility criteria based on title, abstract and descriptor terms. The two reviewing authors scrutinised each article independently, and consensus agreement was reached and documented on all articles meeting the inclusion and exclusion criteria. Disagreements were resolved by the third author (R.T.E.). Relevant literature for the review was subdivided into theme s and critically appraised. Data was extracted from the articles using the Drummond checklist [39] based on study characteristics—country, type of GABS, type of economic approach used, value estimation for accessing and using GABS and recreational activity in GABS, along with health and wellbeing outcomes. Articles were excluded on agreement if there were serious methodological errors, such as applying the quality appraisal process to moderate, rather than exclude, evidence. In addition, the two review authors independently assessed the risk of bias domain into low, moderate and high risks of bias for all included articles. Certainty of and quality of evidence was assessed using the GRADE approach (Grading of Recommendations, Assessment, Development and Evaluation). Articles were selected based on the criteria that the study setting was green/blue, or GABS, applying economic approaches in estimating the value and benefits of accessing and using GABS. Only articles incorporating all components were considered of merit for inclusion in the review. 

## 3. Results

Following the Cochrane systematic review processes [40] and Preferred Reporting Items for Systemic Reviews and Meta-Analyses (PRISMA) guidelines [41], a systematic literature review protocol was developed.

No articles assessed were considered of a high risk of bias; nine studies were considered of good quality, and three studies were considered to be of moderate methodological quality and have valuable outcomes for the review. The characteristics and quality assessment of the included studies are outlined in Table 1 below.

The valuing nature systematic review search concluded with 12 articles meeting the inclusion criteria. Four themes emerged, which included the economic evaluation of green and blue spaces (*n* = 2) and the economic evaluation of green spaces (*n* = 6), which was further subdivided in to urban/built and rural/natural environments, given the perspective of the studies. The third theme was the economic evaluation of blue spaces (*n* = 4), and the fourth theme was the valuation estimates for green and blue spaces for recreational purposes. All articles included in the review involved undertaking recreational activities in the outdoors and evaluated the experiences and associated health benefits.

### 3.1. Subsection

#### 3.1.1. Themes

Theme 1: Economic evaluation of green and blue spacesTheme 2: Economic evaluation of green spacesTheme 3: Economic evaluation of Blue spacesTheme 4: Valuation estimates for green and blue spaces

Theme 1: Economic evaluation of green and blue spaces. Two papers were included in this theme: [43,51]. Clarke et al. (1999) [43] used the stated preferences approach using an interactive computer program to value the preferences for environmental public goods (e.g., wildlife refuge and clean air) or private goods of known market value (e.g., $15 meal and $500 airline ticket) or sums of money ranging from $1 to $9000. Clear air ($720–$737) and wildlife ($700–$711) were the highest-valued in both the societal responsibility and the individual responsibility scenarios. The level of information and priming provided on the social responsibility scenario had an influence on the returning valuations, which impacted on the willingness to pay (WTP). White et al. (2016) [51] used secondary data from a sample of 280,790 adults in the UK between 2009–2015 who reported on their own behaviour of engaging in recreational activities in the environment in the previous 12 months. The methodologies used were a single site’s travel cost model (TCM) and quality adjusted life years (QALYs) to estimate the value of using nature (distance/transport/time spent in the natural environment). Approximately 8.23 million adults made at least one “active visit” to natural environments in the previous week, resulting in 1.23 billion “active visits” annually. An estimated 3.20 million “active visits” reported meeting the recommended PA guidelines (i.e., 150 minutes per week), and active visits were associated with an estimated 109,164 QALYs annually. Assuming the social value of a QALY is £20,000, the annual value of these visits was approximately £2.18 billion, and the implications for health in terms of QALYs was considerable. 

Theme 2: Economic evaluation of green spaces. Six papers were included in this theme, which explored the economic evaluation of green spaces [32,44,45,50,52,53]. The subthemes of urban/built environments and rural/ natural environments were developed from exploration of the literature. 

#### Urban/Built Environments

Urban/built environments are constructed by humans and include buildings, recreational spaces and infrastructure networks such as transportation and utilities. Urban green spaces such as parks are essential for recreational activity, the promotion of PA and improved public health and afford ameliorating health outcomes. A cost analysis study by Wang et al. (2004) [50] explored the construction of five bike and pedestrian trails with costs calculated using discounted rates of 3%, 5% and 10% over 10, 30 and 50 years, respectively, with the average cost per mile per user estimated. Data was collected on user usage on a single day from five trails. Applying the discounted rates to construction costs plus maintenance costs over the specified timeframes and then comparing them with the direct medical cost savings for PA and applying a 5% inflation rate, the savings were $622 in 2002, with cost savings outweighing construction and maintenance costs. The evidence is suggestive that developing trails may be a cost-effective means to promote PA and that future research should examine WTP among the population for infrastructure changes promoting PA, and costs per mile is a useful measure for the estimation. 

The promotion of PA in local communities is increasingly explored to assess if changes in neighbourhood quality could transform health behaviours among the public. The value and demand for walking was examined by Longo et al., 2015 [32] to investigate if changes in neighbourhood characteristics could improve walkability, thus stimulating alterations in health behaviours. Taking a discrete choice experiment (DCE) approach and using compensating variation techniques, data was collected by means of face-to-face interviews from a sample of 1209 respondents over a 12-month period. Econometric analysis used Tobit models to explore the demand for walking estimated by minutes spent walking in local neighbourhoods over a seven-day period, the value of time, along with changes in neighbourhood characteristics and walkability, taking account of respondents’ health status, BMI and demographic characteristics. Estimation of the monetary value of increased minutes walking per week for scenarios of improved neighbourhood quality and walkability suggest that improved perceptions of local areas along with the availability of local amenities had the potential to increase walking on average by 36 minutes per week, with a monetary valuation for walking £13.56 per person per week. Results suggest that a public policy programme that improved walkability has the potential to increase levels of PA by a quarter of the recommended guidelines per week would have an average value per resident of £710 annually. These estimates provide policy-makers with guidance and valuation estimates on potential health behaviour changes to increase the levels of PA and that improvements to local neighbourhoods are valued by the public. 

#### Rural/Natural Environments

Rural/natural environments are predominately open countryside containing natural resources for farming, forestry and leisure, along with the conservation of wildlife and landscapes. The characteristics of environments can impact on the levels of participation in PA and influence health statuses. Applying the measure of health adjusted life years (HALYs), Zapata-Diomedi et al. (2016) [53] explored PA-related improvements in health associated with environmental characteristics linked with cost savings/increases in healthcare costs. The study applied 28 scenarios based on density, diversity, design, destination, distance and walkability of local environments to model HALYs (equivalent to one year of full health due to avoiding an illness and postponement of death) to examine health outcomes associated with undertaking PA in local environments. Results indicate that there are HALYs gained because of quality changes to all environmental characteristics, with walking for transport and the provision of additional recreational destinations seeing potential gains of up to 19.81 HALYs. Healthcare cost savings per year for PA-related diseases ranged from A$1300 to A$105,355 per 100,000 adults. Results suggest there are potential health benefits as well as healthcare cost savings associated with changes in quality of local environment characteristics, promoting PA and opportunistic walking. 

Green spaces such as forests and parks afford the public the opportunity to engage in recreational activities during leisure time, but as these natural resources do not have a market price, the assessment of the use and nonuse values can be difficult to evaluate. To estimate the monetary value of forest recreational services, Jankovska and Straupe’s (2011) [45] research examined the value to Latvia’s national economy of accessing forest recreational amenities and services. Data from Latvia’s state forests and tourism services provided information on the number of people visiting forest sites, along with fees paid and the use of services. In addition, to explore choices and preferences for accessing forests for recreational purposes, data was collected by means of a contingent valuation method (CVM) survey to estimate WTP for access to, along with the value associated with, quality improvements. Results indicate the value to the Latvian economy associated with forests tourism for recreational purposes is 194,000 EUR. The most popular recreational activity was walking in forests and ranking the management of forests maintained in natural environmental conditions with recreational amenities as the preferred option. WTP to contribute to improve amenities in forests were valued at 9.5 EUR per person and 4,362,397 EUR for the entire study sample, indicating the public value of accessing forests and their WTP to improve the quality of amenities at forests for recreational use. Linking forest recreation, health and therapeutic benefits with the economic value of participating in recreational activities in natural environments is gaining increased attention. Willis et al. (2016) [52] investigated the benefits of the forest “branching out” activities for people with severe and enduring mental health problems. Cost-effectiveness analysis (CEA) revealed that the ecotherapy programme is comparable to other programmes oriented to social recovery. Findings indicated improved levels of PA, with costs estimated to be £426 per user and a good value for money in terms of National Institute for Health and Care Excellence (NICE) guidelines.

Doctorman and Boman (2016) [44] used the contingent valuation method to compare health status and WTP to avoid the interruptions of recreational activities in forests among hunters (2500) and forest recreationalists (3000). WTP questions asked about avoiding losing forest recreational experiences, and respondents provided information on their perceived health status through the use of the EQ-5D questionnaire which uses a standardised measure of health-related quality of life, along with BMI and demographic information. Two models were developed: one for hunters and another for forest recreationalists and linked with the frequency of visits. Findings indicated that hunters have a higher WTP threshold and use value to access forests, as well as ascribing a value for perceived changes in health status because of their experience. Hunters had a significantly higher WTP to avoid the postponement of recreational activities (8.87 Swedish Kronor (SEK) compared with forest recreationalists, which was 7.57 SEK). Results also indicated that hunters had a higher marginal WTP (65 SEK) to avoid losing one unit of perceived health status due to a loss in recreational experience, compared with a WTP of 17 SEK for forest recreationalists.

Theme 3: Economic evaluation of blue spaces. Four papers were included in the review that looked at the economic evaluation of blue spaces [46,47,48,54]. 

A policy framework study by Rabinovici et al. (2004) [47] exploring the economic, health and recreational implications of unnecessary closures due to high water contamination levels used a four-stage transfer approach to estimate the value of swimming at the Lake Michigan Freshwater Beach. The study examined the introduction of a water quality testing programme at a cost of $250 per day and the value transfer parameters that visitors attribute to health and recreation at the lake. Results for the transfer constraints and based on visitor numbers indicated the value per visitor per day of swimming recreation was $16.02 on low attendance days and $38.46 on high visitor attendance days. The examination of the value transfer for the value per day of avoiding health effects of poor-quality water was estimated at mild ($280) and moderate at $1125 per visitor per day. Results indicate a typical closure causes a net economic loss among would-be swimmers totalling $1274–$37,030 per day, depending on the value assumptions used.

Frequently, the supposition is that, for recreationists’ costs of traveling to a site, a reliable measure of the value placed on that resource and the recreation opportunities is provided. A TCM survey by Smith and Moore (2012) [54] sampled 247 respondents to examine the demand for recreational activities at two rivers. TCM results indicated a variance in the average number of visits to both rivers, with the frequency of visitations associated with the proximity and length of travel distance. On average, the cost to visit a river for recreational activities varied from $128 to $393 and directly linked with travel distance, length of time spent at the recreational site and an individual’s affective and emotional attachments to recreation settings, with experiences influencing the recreation demand. 

To value the environment quality, a discrete choice experiment (DCE) study by Remoundou et al. (2014) [48] exploring choice and preferences was conducted to investigate the impact of changes in environmental management strategies along with water quality, biodiversity and level of health risk influencing valuation contributions towards proposed marine protection programmes. Two questionnaires were developed differing only in public good, budget and tax reallocation of funding within public budgets. A model based on the willingness to reallocate (WRA), WTP and Marginal Rate of Substitution (MRS) was developed and derived as equivalent and dependent on the value of disposable income when compared with quality changes and alternatives. Results indicate that respondents were willing to redirect money to the introduction of marine protection programmes to reduce the level of public health risk and improve the level of water quality and improve marine biodiversity. 

Linking quality improvements and health benefits, a study used data from the Health Survey for England (HSE) [46] from 10,333 private households to compare alternative marine spatial plans. Results estimated that physical activities undertaken in aquatic environments at a national level provided a total gain of 24,853 QALYs. A conservative estimate of the monetary value of a QALY gain of this magnitude was £176 million. The approach provided insight on redirecting funds towards recreational facilities and ensuring society can easily access and enjoy the natural environment adequately considered at the local, regional and national levels. 

Theme 4: Valuation estimates for green and blue spaces. The literature examining the value of ecosystems in momentary terms focused on forest parks and urban green spaces. Forestry and river evaluations demonstrated the values of the individual and the economy [45] beneficial impacts on mental health [52], wellbeing effects of recreational activity [46,51,52] and local urban green spaces facilitated improvements in physical health [32].

To examine the choices and preferences, along with the estimated value of accessing and using GABS, the included studies predominately used primary data collection, with one study using secondary data [29]. Stated preference (SP) [32,43,44,50,51] and revealed preference (RP) [51,54] techniques or combined approaches [45] were applied in the selected studies. These methodological approaches focused on developing a better understanding of choices and preferences among populations, with the task of estimating monetary valuations for nonmarket goods and associated decision-making mainly derived from techniques applied in environmental economics translated to health outcomes and value estimates. The value estimates that the public places for accessing and using GABS are outlined in Table 2. Valuations varied across timelines, as well as in currency and monetary valuations, given the heterogeneity of the included studies with regards to GABS settings, the economic evaluation approach used population and health and wellbeing outcomes. To present results in consistent monetary denominations, inflation and currency conversion calculators were applied for each of the studies monetary valuations. All WTP estimates are presented in local currency, as well as GBP £, Euros and US $. Findings indicated that the public are WTP between £5.72 [44] and £15.64 [32] in 2019 value estimates for not postponing or losing the health benefits of an outdoor experience, as well as the value associated with walking in local environments. The monetary estimations demonstrate the value the public allocated to accessing and using GABS under current and enhanced environments to improve their health and wellbeing outcomes.

## 4. Discussion

This systematic review explored the economic evidence regarding the costs of undertaking PA in GABS worldwide. Twelve papers were included and categorised into the themes of green and blue spaces, green spaces and blue spaces. As the number of studies was low and the subject matter quite varied, no meta-analysis was possible due to the unique nature of all the studies. The economic evaluation methods used included a paired comparison technique [43], quality adjusted life years (QALYs) [46], cost-effective analysis (CEA) [52], health adjusted life years (HALYs) [53], contingent valuation method (CVM) [44,45], costs and visitor numbers [50], travel cost model (TCM) [50,51,54], discrete choice experiment (DCE) [32,48] and a policy framework [47].

The application of the various economic methodologies allowed for the exploration of various quality improvements to estimate the public’s value for access to GABS, as well as opportunities to increase levels of PA. The evidence indicates that the public are WTP for quality improvements to increase opportunistic walking and associated health benefits [53], as well as building purposeful walking into daily activities and chores [32]. The investigation of access to natural environments for recreational opportunities and valuation estimates suggest that the public value access to recreational opportunities to increase levels of PA and are unwilling to forgo the opportunity [44].

Valuation estimates are presented in more than one currency domination so that the value of GABS can be applied across multiple countries. The valuation estimates when extrapolated forward by means of the inflation and currency conversion calculators allowed for current estimates to be calculated for the use and nonuse values of accessing and undertaking recreational opportunities in GABS and applied to six of the papers reviewed [32,44,45,50,52,54]. The valuation estimates in 2019 across multiple dominations indicate that the public value leisure opportunities in natural environments and are WTP not to forgo the opportunity costs and health benefits lost to relinquishing engaging with natural environments and undertaking recreational activities [44]. Opportunity cost is defined as the next best alternative forgone to satisfy the particular wants, and the central economic problem is the scarcity of resources to satisfy unlimited wants and needs. Due to scarcity, societies need to find a way to allocate resources to generate maximum benefits, and the public values walking in local environments [32], with leisure activities contributing substantially to local economies [45,47], and the public are willing to incur the travel costs to visit and undertake recreational activities in natural environments [54]. 

When considering future policy developments for GABS and sustainable development, the evidence to date recommends that policy-makers should consider the following; creating better walkable environments [32]; take account of the land use, spatial planning and planning considerations [46,47]; health and wellbeing outcomes for the public [10] and incorporating green spaces into mental health interventions in improving mental health outcomes [52]. Recommendations for future policies and research should take a more integrated multisystem approach to public health and be inclusive of local and spatial authority planning and meet the needs of transport and natural resources to take account of the impact and values of local environments to exploit health benefits and outcomes for the public.

## 5. Conclusions

This systemic review is the first assessment in the health economics literature in this field evaluating natural environments and health. The economic evaluation methodologies used approaches mainly applied in environmental economics, which have been adapted to include and reflect health and wellbeing changes and valuation estimates. This review shows evidence on the ability to transfer and apply varying methodological approaches across disciplines effectively in addressing complex multifactorial issues regarding the investment and allocation of resources promoting health-improving behaviours. In conclusion, scrutiny of the evidence on natural environments, including GABS, provided the context for a large proportion of recreational PA. There is a need to protect and manage such environments for health purposes [51], as well as maintaining ecosystems and flora and fauna [43]. The use of natural environments for PA allows the public to meet the recommended guidelines for PA and associated health benefits. Evidence suggests that there are health benefits associated with recreational activity in green spaces and that the public perceives there are health gains and are WTP to access environments and contribute to quality improvements to engage in PA in green spaces. Blue spaces are valued by the public; however, it is also clear that more research is needed to create empirical and theoretically more robust estimates of blue space recreation demands and quality of life benefits. 

## Figures and Tables

**Figure 1 ijerph-17-04142-f001:**
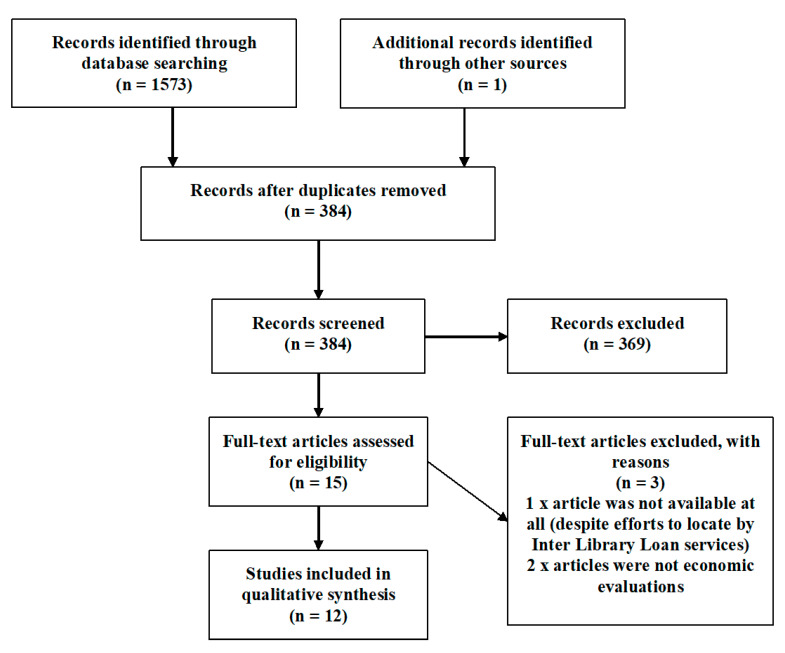
Preferred Reporting Items for Systemic Reviews and Meta-Analyses (PRISMA) 2009 [42] flow diagram for the valuing nature systematic review.

**Table 1 ijerph-17-04142-t001:** Characteristics and quality assessment of the included articles.

Study Reference (Author, Year)	Country	Type of Nature Based Study	Economic Method	Quality Assessment of Economic Evaluation Method
Clarke et al. (1999) [43]	United States of America (USA)	Environmental public goods	Paired comparison techniques	Moderate
Doctorman and Boman (2016) [44]	Sweden	Forest recreation	Contingent Valuation Method	Low
Jankovska and Straupe (2011) [45]	Latvia	Forest recreation	Travel Cost Model and Contingent Valuation Method	Low
Longo et al. (2015) [32]	United Kingdom (UK)	Improved walking infrastructure	Discrete Choice Experiment	Low
Papathanasopoulou et al. (2016) [46]	United Kingdom (UK)	Physical activity in marine environments	Quality Adjusted Life Years (QALYs)	Low
Rabinovici et al. (2004) [47]	United States of America (USA)	Lake recreation	A policy framework	Moderate
Remoundou et al. (2014) [48]	Greece	Marine restoration and public funding	Discrete Choice Experiment	Low
Smith and Moore (2012) [49]	United States of America (USA)	River recreation	Travel Cost Model	Low
Wang et al. (2004) [50]	United States of America (USA)	Recreational trails	Cost analysis	Moderate
White et al. (2016) [51]	United Kingdom (UK)	Recreational activities in the environment	Travel Cost Model with QALY ratios	Low
Willis et al., 2016 [52]	United Kingdom (UK)	Forest recreation	Cost-Effective Analysis (CEA)	Low
Zapata-Diomedi et al. (2016) [53]	Australia	Physical activity in local environments	Health Adjusted Life Years (HALYs) models	Low

**Table 2 ijerph-17-04142-t002:** Monetary valuations for green and blue spaces. WTP: willingness to pay.

Author and Year of Study	Value in Year of Study	Year 2019	GBP£ 2019	Euro 2019	US $ 2019
**Willis et al., 2016** [52]					
Cost per user of the ‘Branching Out’ programme	£426	£464.71	£464.71	€524.54	$566.78
**Doctorman and Boman (2016)** [44]					
WTP to avoid postponement of recreational activities					
Hunters	8.87 SEK	9.51 SEK	£0.79	€0.89	$0.97
Forest recreationalists	7.57 SEK	8.45 SEK	£0.70	€0.79	$0.86
WTP for not losing the health benefits of outdoor experience					
Hunters	65 SEK	68.65 SEK	£5.72	€6.45	$6.97
Forest recreationalists	17 SEK	17.96 SEK	£1.50	€1.69	$1.82
**Wang et al. (2004)** [50]					
5% discount rate over 30 years	$83.00–$592.00	$112.33–$801.21	£92.13–£657.13	€103.92–€741.21	$112.33–$801.21
Construction costs per mile	$5725–$45,505 (2002)	$8135.83–$64,667.44	£6672.01–£53,032.28	€7692.42–€59,819.55	$8135.83–$64,667.44
5% inflation rate re: savings	$622.00	$883.93	£724.89	€817.67	$883.93
**Longo et al. (2015)** [32]					
Value of walking in local environments	£13.56	£15.64	£15.64	€17.64	$19.08
**Jankovska & Straupe (2011)** [45]					
Benefit to Latvian economy	€194.00	€213.92	£189.58	€213.92	$231.28
Individual WTP to improve environment	€9.50	€11.03	£9.77	€11.03	$11.93
Entire sample WTP to improve the environment	€4,362,397.00	€4,810,379.48	£4,262,537.55	€4,810,379.48	$5,200,733.01
**Smith and Moore (2012)** [49]					
Average cost to visit a river	$128–$393	$138.07–$423.91	£122.33–£375.58	€127.72–€392.14	$138.07–$423.91

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
