# Peer review of "A Systematic Review Exploring the Economic Valuation of Accessing and Using Green and Blue Spaces to Improve Public Health"

_ijerph, 2020, doi:10.3390/ijerph17114142_

Round 1

Reviewer 1 Report

I think the paper is well written and organized overall. In fact, it was my first time reviewing a non-empirical paper but it was quite interesting and I feel like the paper like this could be beneficial to the field of public health as well as general public.

However, there are so many acronyms used in the article, although most of them are explained, and it would be great to have a table or glossary in my opinion. I had to go back to the previous page several times in order to figure out some acronyms while moving forward. I believe "GP" on the page 2 stands for a "General Practitioner" by the context, but it wasn't explained in the paper (probably because everyone knows it in the UK) and it is not a popular term in the United States for sure.

Other than that, the paper is well analyzed and organized in my opinion and it could be a guideline for the professionals in public health field.      

Author Response

Thank you for reviewing our paper and for your advice and guidance on revisions to improve our re-submission. These suggestions are gratefully appreciated. We have now included a glossary of terms as Appendix 1. 

Reviewer 2 Report

This systematic review synthesizes evidence related to the economic value of green and blue space. The methods are appropriate and described clearly and transparently. The findings are well summarized. Overall, this study is well designed and implemented, fills a critical gap in the literature, and has important policy implications. I have a few suggestions that hopefully will help the authors improve the manuscript.

Title

I suggest the authors consider highlighting the “economic valuation” concept in the title, as this is the unique contribution of this research.

Introduction

The authors mentioned some benefits of GABS that can be translated into economic terms. However, the significance of assessing the economic value of these benefits is not adequately explained. A significant barrier to developing policy on green and blue space is that the benefits are long term and hard to assess. There is a critical need to bring to light the economic gains of such strategies. It would be a missed opportunity not to highlight this need.

 Method

I find the Method section clearly written and easy to follow.

It is helpful to include the month-year when the search was conducted to provide context for the search results.

The author should consider listing the three groups of search terms either in the manuscript or as an appendix.

Results

It would be very interesting to summarize the different types of health or ecosystem services that are evaluated in the literature. For example, some articles include valuations of clear air/water; some mainly focus on valuations of PA. A synthesis of the mechanisms of benefits that have been discussed and the types of benefits that remained unknown in economic terms would be an important contribution.

The results are a bit term-heavy. For example, in Line 177, “social responsibility” and “sole responsibility” scenarios are a bit hard to understand. Considering the audiences have not read the original papers, it would be helpful to paraphrase some of the terms used in the original articles to aid understanding.

Discussion

The Discussion could benefit from some perspectives about the big picture. The authors can include some discussions related to the policy implications of this study. In addition, it would be helpful for the authors to point out research gaps and directions. What recommendations can the authors provide for researchers interested in conducting studies in this field?

Author Response

Thank you for reviewing our paper and the valuable suggestions your provided. We have changed the title to include ' economic valuation' as suggested. 

In the introduction we have added additional content outlining the economic value and estimates of accessing and using parks and green spaces and recommended policy changes for walkable environments.

Date of key word search is now included which was January 2018.

Search terms are included in Appendix 3 and reference to the three columns of search terms is in text.

Words changed to to 'societal' and 'individual' choices for clarity. 

Theme 4- value estimations new content at the beginning of the section to summarise the ecosystems associated with the valuation estimates and the health and well-being benefits.   

New paragraph added to the discussion section which takes account of policy and linked with recommendations for future policy development and research.

Reviewer 3 Report

The topic of this paper is very pertinent. The aim of the research was to explore the economic evidence associated with the public’s value for accessing, using green or/and blues public spaces in relation to public health.

The authors should mention only the value to public health and should not mention the other ecosystem services provided by access to green infrastructure. The text should be focused to health and not mention benefits to wildlife as in line 176 wildlife ($700-$711).

At the group of words used for this systematic literature review is missing the word beach or coast, which is the main blue public space related to health, with about a third of the world tourism to be at a beach and majority of health benefits from visiting a blue space. The word "Sea" is related to cruise tourism or sailing, which are beneficial to health but are activities for a minority of users with a lot of money. Taking sunbaths at the beach is an activity that improve health.

Another type of blue space that was totally ignored was the thermal waters, which are very important in eastern Europe and recently in new markets like Island, Hungary. 

In green public spaces it was not mentioned the impact to improve the urban microclimate and decrease the heat island, protecting from the consequences of weatwaves. 

Benefits to health from green spaces filtering the air, especially in heavily polluted urban areas was barely mentioned. Also benefits from noise reduction. The people can do exercise activities in many public not green spaces (like biking along the streets) but only in green spaces they can exercise with cleaner air.

Another big group of papers related to health benefits is the view from the window. Maybe the authors want to mention only the physical access to green/blue spaces but even the direct view has health benefits, (in some cases was presented empirical data with 30% decrease in number of days needed to recover from surgery when having view to green space). 

Also it was not mentioned any hedonic valuation method. There are many studies about higher valuation of properties or hotels when are close or with view to green/blue spaces and this is due to health and wellbeing benefits. 

Finally I want to mention that the willingness to pay is related to demand and offer. The people from rural areas have a lot of green space and are not willing to pay for health benefits, meanwhile this does not mean that they do not recognise the value of doing activities and all their life in green spaces. 

The value of WTP 5.72 at the abstract might be misleading. It is written in a way that means that the people accept to pay a fee for their visit to public green/blue  space. This is not truth for majority of countries that consider that public space should be free for access. WTP should not be confused with willingness to spend, or willingness that the state should spend our taxes for green space conservation. There are countries like USA and UK that visitors respond positively in their WTP fees and this is related to the services they are used to get and not related to the value they give to benefits to health. The abstract should mention the self-perceived health benefits from the public and the other methods.

The keywords after the abstract, should not repeat words from the title of this paper. 

Author Response

Econo

Response 1:

Under Theme 1: Economic evaluation of green and blue spaces, wildlife is highlighted as it has an economic value and is an environmental public good which has an associated momentary valuation.  This was evaluated within the reviewed literature for this systematic review and the included article used stated preferences approach using interactive computer program to value preferences for environmental public goods, Clarke et al., (1999) [45]. Clear air ($720-$737) and wildlife ($700-$711) were the highest valued in both the societal responsibility and the individual responsibility scenario.

Response 2:

In the review search terms words used were specifically to describe actual water mass which was considered blue space not the associated surrounding area.

Response 3

As this paper is a systematic review exploring the economic valuation of accessing and using green and blue spaces to improve public health the focus of the search was to integrate the evidence which incorporated economic valuation techniques to estimate the value of the health benefits of public environmental goods or services.

Although the points made are pertinent regarding the impact of environmental impact of climate change on both urban and rural areas this environmental aspect was not a focus of this systematic review. The hedonist valuation method is associated with property valuations and used within environmental economics the methods was not used in any of the studies meeting the inclusion criteria for this review. However, other well established environmental economic techniques such as TCM, CVM and DCE were used within the studies reviewed to reflect the valuation estimates.

Response 4:

Following your advice, we have revised the wording and content which is outlined below to take account of the variations in WTP estimates associated with accessing and using green and blue spaces within the abstract:

Results suggest the public value access to green and blue spaces to undertake recreational activities and avoid delay or losing the recreational experience and associated health benefits. The public are Willing To Pay between £5.72 and £15.64 in 2019 value estimates, for not postponing or losing an outdoor experience and for walking in local environments under current and improved environmental conditions, respectively. Valuation estimates indicate the public value green and blue spaces and are willing to pay to improve local environments to gain the health benefits of undertaking leisure activities in green and blue spaces.

The following text has been added to the paper and indicates that valuations varied by country, currency as well as timelines. Lines 335 to 340 in main body of text:

Valuations varied across timelines as well as in currency and monetary valuations given the heterogeneity of the included studies in regard to GABS settings, economic evaluation approach used, population and health and wellbeing outcomes. To present results in consistent monetary denominations inflation and currency conversion calculators were applied for each of the studies monetary valuations. All WTP estimates are presented in local currency as well as GBP £, Euro’s and US $.

Response 5: Key words are words or phrases that capture the most important aspects of an article. Therefore, the use of the included key words captures the essence of the article and can not be improved upon.

Reviewer 4 Report

This is an excellently written paper addressing one of the largest gaps in this field of research i.e. economic valuation of green/blue space benefits.  This is be a very valuable contribution to the field.  Use of systematic search criteria and exploration are excellent, as are comparison of different WTP methods across the analyzed studies. 

Two slight but important adjustments are recommended:

  1. Strongly recommend adding the term “economic” value to the title of your paper. This is a topic of great importance and limited availability in this field, and most papers on this topic investigate health-based “value” (e.g. clinical efficacy, etc…).  Explicitly stating your paper is exploring the economic value of green/blue space will notify the reader directly about the purpose of the paper as well as increase its detection via Search functions for people looking for information on this important and under-explored topic. 
  2. WTP is identified as up to £5.72 in the abstract, but it is unclear in the body of the manuscript how this amount was calculated. The only other occurrence of this amount is in referencing Doctorman and Boman’s paper [46].   Please send more time developing and explaining the calculations cited in your abstract for this central part of your paper. 

Only other comment is the recommendation to put in a paragraph break in the introduction after reference [27].  It makes the content more readable and visually accessible to the reader. 

Author Response

Response 1: 

Thank you for reviewing our paper and providing valuable feedback. In taking account of this we have changed the title to:

A systematic review exploring the economic valuation of accessing and using green and blue spaces to improve public health.

Response 2:

The abstract has been changed to the following for clarity:

The public are Willing To Pay between £5.72 and £15.64 in 2019 value estimates, for not postponing or losing an outdoor experience and for walking in local environments under current and improved environmental conditions, respectively.

In addition, under Theme 4: Valuation estimates for green and blue spaces content revised to the following:

The value estimates that the public place for accessing and using GABS are outlined in Table 2. Valuations varied across timelines as well as in currency and monetary valuations given the heterogeneity of the included studies in regard to GABS settings, economic evaluation approach used, population and health and wellbeing outcomes. To present results in consistent monetary denominations inflation and currency conversion calculators were applied for each of the studies monetary valuations. All WTP estimates are presented in local currency as well as GBP £, Euro’s and US $. Findings indicated that the public are WTP between £5.72 [46] and £15.64 [35] in 2019 value estimates, for not postponing or losing the health benefits of an outdoor experience, as well as the value associated with walking in local environments. The monetary estimations demonstrate the value the public allocate to accessing and using GABS under current and enhanced environments to improve their health and wellbeing outcomes.

Response 3:

Paragraph break inserted after reference 27.

Round 2

Reviewer 3 Report

The paper was significantly improved. The authors should rethink their keywords.